# Reproducibility and Repeatability Tests on (SnTiNb)O_2_ Sensors in Detecting ppm-Concentrations of CO and Up to 40% of Humidity: A Statistical Approach

**DOI:** 10.3390/s23041983

**Published:** 2023-02-10

**Authors:** Michele Astolfi, Giorgio Rispoli, Sandro Gherardi, Giulia Zonta, Cesare Malagù

**Affiliations:** 1Department of Physics and Earth Sciences (UNIFE), Via Saragat 1, 44124 Ferrara, Italy; 2SCENT S.r.l., Via Quadrifoglio 11, 44124 Ferrara, Italy; 3Department of Neuroscience and Rehabilitation, Via Luigi Borsari 46, 44121 Ferrara, Italy

**Keywords:** chemoresistive sensors, metal oxide, volatile compounds, sensor response, repeatability, reproducibility

## Abstract

Nowadays, most medical-diagnostic, environmental monitoring, etc. devices employ sensors whose fabrication reproducibility and response repeatability assessment are crucial. The former consists of large-scale sensor manufacture through a standardized process with almost identical morphology and behavior, while the latter consists of giving the same response upon repeating the same stimulus. The thermo-activated chemoresistive sensors, which change their conductance by interacting with the molecules composing the surrounding gas, are currently employed in many devices: in particular, thick-film (SnTiNb)O_2_ nanosensors were demonstrated to be particularly suitable in the medical and biological fields. Therefore, a set of thirteen of them, randomly selected from the same screen-printing deposition, were laboratory tested, and the outcomes were statistically analyzed in order to assess their consistency. At first, the working temperature that maximized both the sensor sensitivity and response repeatability was identified. Then, the sensors were subjected to different gas concentrations and humidities at this optimal working temperature. It resulted in the (SnTiNb)O_2_ nanosensors detecting and discriminating CO concentrations as low as 1 ppm and at high humidity degrees (up to 40%) with high repeatability since the response relative standard error ranged from 0.8 to 3.3% for CO and from 3.6 to 5.4% for water vapor.

## 1. Introduction

### 1.1. MOX Sensors Overview

Metal-oxide (MOX) sensors are widely used for gas detection and employed in many application fields, such as automotive, environmental monitoring, agri-food, and diagnostic, thanks to their cheapness, high reliability, and great versatility. They present many advantages with respect to other gas sensor types (as optical, acoustic, catalytic, etc.) because of their ease-of-use, high sensitivity (detecting concentrations of specific gases as low as few parts per billion; ppb), low detection limit, low recovery and response times (up to some minutes), and long duration (several years) without any maintenance. However, their poor selectivity and their performance worsening in the presence of interfering humidity are their highest limits. Nevertheless, the former can be overcome by combining them in an array, and the latter by mixing different MOX materials in order to minimize the water vapor influence with the sensor active material [1,2]. In order to obtain high-quality MOX sensors, their manufacture process requires several delicate and specific procedures. Indeed, it is crucial to increase as much as possible the surface–volume ratio of their sensitive porous material to maximize its interaction with the surrounding gases [1,2]. With this aim, the sol-gel chemical process was employed to synthesize networks of MOX nanometric spherical grains (with a size ranging between 50–200 nm), precisely interconnected, and endowed with the appropriated doping with oxygen vacancies (working as electron acceptors) [3,4,5]. The sol-gel method presents many strength points to synthesize nanostructured materials: besides the high costs of the raw starting materials, it allows us to obtain very stable, homogeneous, and pure products with low temperature treatments and negligible processing costs, with a full control on the final material thickness and porosity [6,7]. In order to reliably employ these sensors in many different applications, especially in medical diagnostic devices (for instance to detect and monitor pathologies as cancer), their production process must be standardized so to have a highly reproducible morphology and behavior (reproducibility) [8,9,10]. Moreover, the produced sensors must give consistent responses upon repeating their exposition to the same gases, even at concentrations of few ppm (repeatability) [11,12].

### 1.2. MOX Sensors Reproducibility and Repeatability Assessment

In this paper, thirteen thermo-activated thick-film (SnTiNb)O_2_ sensors (named STN in the following), whose sensing material was obtained through a standardized sol-gel synthesis and applied on the sensor substrate by using a screen printer (see Materials and Methods) [3], were randomly selected from the same printing batch and concurrently tested with different concentrations of CO and water vapor (the latter was used as interfering and target gases separately [12,13]). In general, thick-film technology is used to produce components manufactured with films having a thickness between 1 and 100 μm by an additive process involving the deposition of several conductive, resistive, and/or dielectric layers onto a substrate usually using a lithographic process. In particular, the thick-film sensors used here are characterized by a film of 20–30 μm of thickness, including several layers of nanostructured material (see Materials and Methods). The outcomes were then thoroughly analyzed with several statistical methods in order to evaluate their reproducibility and repeatability and also under unfavorable conditions (high water vapor concentrations and/or low gas concentrations). This sensor type has been demonstrated to be particularly suitable to detect and monitor pathologies as cancer in several previous works [14,15,16,17]. Therefore, it has been examined in this paper, for the first time, if their production process is standardized enough to have a highly reproducible morphology and behavior, by statistically testing a randomly selected pool of thirteen sensors. To this aim, these sensors were tested at first in a CO atmosphere at different working temperatures (WT) to choose the optimal one (i.e., the temperature at which the sensors gave the highest, most repeatable, and most reproducible responses) [18,19]. Then, the sensors, heated at the optimal temperature, were subjected to different CO concentrations (from 1 to 10 ppm, according to the gas concentration exhaled, for example, by biological samples) and humidity percentages. Since this sensor type proved to be particularly suitable in medical screening devices (developed by our laboratory in the last years), to analyze the tumoral volatile organic compounds exhaled by various biological samples [14,15,16,17,20], in particular by blood samples [15,20] and tumoral cell cultures [17], it is crucial to guarantee their reliability in time through a statistical approach. Moreover, STN sensors proved to be very stable since they keep giving consistent responses to the above biological samples for more than three years. However, these sensors share the poor selectivity typical of the MOX sensors, therefore they must be used in an array of different ones in order to improve the selectivity to a certain target gas. Therefore, to assess the sensor production reproducibility, outcomes repeatability, and to monitor their behavior, also in unfavorable environmental conditions, a pool of thirteen STN sensors, obtained by the same production batch and randomly selected, were thoroughly laboratory tested and their outcomes were statistically compared. Further measurements were performed on Dulbecco Modified Eagle Medium samples as well in order to evaluate the repeatability of the sensor results in the presence of volatile organic compounds and humidity percentages typical of biological samples.

## 2. Materials and Methods

### 2.1. Synthesis and Film Deposition

(Sn, Ti)O_2_ solid-solution (Sn:Ti = 7:3) was obtained by the sol-gel technique from stoichiometric solutions of their metal-alkoxides precursors (Sn(II)-ethyl hexanoate 95% Aldrich, Ti-butoxide 97% Aldrich) [3]. The metalorganic molecules were hydrolyzed by dropping a diluted HNO_3_ solution to the (Sn_0.7_Ti_0.3_)O_2_ one, while gently stirring the solution with a magnetic anchor; the appearance of a pale yellow color indicated that the colloid solution was ready to work as a solid solution precursor. After the filtration of the colloid material, the resulting xerogel was transformed into a binary oxide by calcining at 550 °C for 2 h under air in a muffle oven. This process lead to an almost (84.5%) pure white (Sn_0.7_Ti_0.3_)O_2_.

By using the sol-gel technique, Nb(V) was inserted into (Sn_0.7_Ti_0.3_)O_2_ by dropping NbBr_5_ (Sn:Nb = 20:1), previously dissolved in a minimal amount of methanol, in the Sn(II)-Ti(IV) stoichiometric solution. The further hydrolysis of all these materials (by using diluted HNO_3_) led to a white colloidal precursor; the following filtration and calcination (550 °C for 2 h in air) of the precursor held to a high purity, 85.5%, (Sn, Ti, Nb)O_2_, (STN in the following).

The STN paste was then deposited on the sensor substrate in form of a thick film (~30 μm) by using a screen-printing machine (Aurel C920, Modigliana (FC), Italy); the printing process is largely employed because of its low cost, fast preparation time, and deposition homogeneity [21,22].

Finally, the thick films were hardened, and all the impurities were removed by firing the deposited paste at 650 °C in an oven.

The grain size homogeneity and the porosity of the sensor sensing film was assessed employing the scanning electron microscope (SEM) technique (EVO 40, Zeiss, Oberkochen, Germany) up to 20.000× of magnitude.

### 2.2. Sensors Structure

The STN sensors produced for this work were entirely assembled at the Sensor Laboratory of the Department of Physics and Earth Sciences of the University of Ferrara. Sensor active material consists of a nanostructured semiconductor thick (~30 μm) film in the shape of nanograins, capable of detecting gaseous chemicals at low concentrations (theoretically up to tens of ppb). Sensors are typically composed of three independent parts: (i) the substrate, (ii) the sensing film, (iii) and the heater (Figure 1).

The substrate (composed of sintered alumina) has the double function of an insulating layer and a mechanical support for the sensor. On its top, it hosts two interdigitated comb-shaped gold contacts, connecting the sensor to its signal transduction circuit. The sensing film (1 mm × 1 mm) was screen-printed on the substrate top (see Section 2.1), electrically connecting the two gold electrodes. A platinum coil was placed underneath the substrate to heat the sensor to its proper WT, fundamental to activate its sensing layer. The sensor heating is adjusted finely by controlling the current flowing through the coil with dedicated external electronics. The WT was calculated from the heater voltage and the current flowing through it [15,16,17]. The sensor heater and the film electrodes were then connected by thermo-compression to a 4-pins TO-39 socket using golden wires (Ø ≈ 0.06 mm), by means of a bonding machine. Hence, each sensor was connected, through the above pins, to dedicated electronics consisting of a signal transduction circuit based on an inverting operational amplifier, which converts the film resistivity changes to a voltage variation. The sensor response (R) was then calculated as:(1)$$R=\frac{V_{gas}}{V_{air}},\;$$
where $V_{gas}$ and $V_{air}$ are the steady state sensor signal in target gas presence and in dry air only, respectively. Equation (1) converts the sensor signal voltage in a dimensionless quantity, called response, thus untying the sensor outcomes from the measured physical quantity (i.e., resistance, conductance, voltage, etc.). In other words, the formula normalizes the sensor signal in the presence of gas over the baseline value.

### 2.3. Experimental Set-Up

STN sensors have been placed inside a hermetically sealed aluminum chamber (Figure 2), able to host and simultaneously test up to eight sensors under different conditions of temperature, humidity, airflow, and gas concentration. The temperature and humidity (crucial parameters to optimize the sensor performances) were monitored by means of a dedicated sensor, placed in the center of the chamber. Synthetic dry air (80% N_2_ and 20% O_2_) and ultra-pure CO (99.99%) technical gas were withdrawn from gas cylinders and diffused uniformly starting from the chamber center (Figure 2).

To have a desired humidity percentage in the sensor chamber, the carrier gas was bubbled into water by means of a specific set-up and monitored by a dedicated humidity sensor. In order to minimize possible temperature changes, the sensor chamber was placed inside a climatic one (Angelantoni CT200, Perugia, Italy), where the temperature, humidity and air composition were kept constant.

A computer-controlled mass-flow regulator, by means of a dedicated software, routed the desired gas at the chosen concentrations (from 0 to 100 ppm) to the sensor chamber through Teflon pipelines.

## 3. Results

### 3.1. SEM Investigation of the Sensor Film

In order to investigate the homogeneity, the porosity and the grain interconnection of the sensor film surface, the thirteen sensors were imaged by means of a scanning electron microscope (SEM) [23], following the laboratory tests. Since the film resistance is very high at room temperature (up to several GΩ), the sensors were gold plated (through sputtering technique): the resulting film conductivity increase led to an improved SEM image resolution. At the highest magnification (20.000×), the porosity motif of the sensor film became evident, showing clusters of grains whose size ranged from 50 to 300 nm. Since the sensor images appeared very similar in grain-grain interconnections and in grain size, just a representative sensor is shown in Figure 3 at four different magnifications, Working Distances (WDs), and Electron High Tensions (EHTs).

### 3.2. Best Working Temperature Selection and CO and Humidity Detection

As first, it is necessary to find out the optimal WT for the thirteen sensors, at which R is maximized and shows the best repeatability. Indeed, WT strongly depends on several parameters such as the film thickness, film morphology, sensing material type, and so on [24]. With this aim, the first step is to find out how the sensor conductance changes with temperature. The mechanism generating this phenomenon can be pictured by the Arrhenius plot (Figure 4), representing the sensor film conductance (G) versus the reciprocal of the absolute temperature (T), here in dry air. This plot identifies the following three regions: in the first one, between 2.8 and 2.0 (1000∕°K), the conductance increases in an almost linear fashion upon increasing T. This relationship originates from the progressive increase of the number of conduction band electrons, despite the incomplete ionization of the environmental gas that rarely interacts with the sensor surface. In the second region (in between 2.0 and 1.5), the conductance is roughly constant despite the temperature increase because the further increase of the number of conduction band electrons (due to thermionic effect) is compensated by an almost equal number of electrons captured by the ionized gas absorbed on the sensor surface. In other words, in this region, the grain-grain potential barriers do not change with the temperature. Finally, in the third region (in between 1.5 and 1) the gas surrounding the sensor is completely ionized and covers almost completely the sensor surface. Therefore, the contribution of the gas in seizing the electrons from conduction band is outrun by the one due to the thermionic excitation of electrons, leading to a decrease of the grain-grain potential barrier. In order to understand if the sensor activation occurs in nearly the same conditions and whether they behave similarly during the gas-sensor film interaction, all the thirteen sensors were statistically analyzed with the Arrhenius Plot. For clarity, the five most diverse Arrhenius plots out of the thirteen ones have been compared in the same graph (Figure 4), and then all of them were averaged and plotted (blue line) in Figure 5 together with their positive (green line) and negative (red line) standard error curves. The almost negligible standard errors confirmed that all the thirteen sensors behaved, chemically and physically, very similarly as expected.

Once ensured the strong similarity of the Arrhenius plots, the thirteen sensors were concurrently tested by exposing them to 10 ppm of CO. The sensors were heated to different increasing temperatures (from 300 °C to 550 °C, with discrete jumps of 50 °C), and for each temperature, they were exposed to CO thrice.

The average value and the standard errors of all the STN sensor responses were calculated for all the WTs tested (Table 1, Figure 6). The optimal WT, giving the highest response and smallest standard error (R=2.31±0.03), was found to be 500 °C (light red candle in Figure 6). This WT was therefore employed for all the following tests.

The sensors, heated at the optimal WT of 500 °C (Section 3.1), were subjected to different CO concentration steps (10, 8, 6, 4, 2, and 1 ppm; progressively lightening the blue bars in Figure 7) to assess their response sensitivity and resolution. In Figure 7 the responses of five representative sensors are plotted as a function of time: the sensors are exposed to each CO concentration for ten minutes, and they recovered their baseline within thirty minutes from the last CO exposition (1 ppm); the low detection limit was found to be about 0.2 ppm.

Since the MOX sensor responses are, in general, rather affected by humidity [12,13,24,25,26,27,28], the above CO concentration steps were performed at first in dry conditions and later in the presence of different relative humidities (RH: 10%, 15%, and 30%; Figure 8); the sensor chamber environment was therefore stabilized at the desired RH for at least six hours before the CO injection. The sensor measurements, in the presence of different CO concentrations and RHs, were repeated three times in order to assess the repeatability of their responses. As expected, the responses increased as RH decreased [29,30], showing the highest values in dry conditions and the lowest ones at RH = 30%. Moreover, the STN responses increased along with CO concentration at all RH conditions tested.

To further calibrate the sensors, it is crucial to assess their behavior in the presence of a progressively increasing concentration of water vapor added to dry air. Each one of the thirteen sensors was subjected to a fixed RH (ranging from 0 to 40% in 10 steps, lasting ten minutes each) for three times (to assess the measurement repeatability) and these three responses, after subtracting the baseline (i.e., zero response corresponds to zero humidity), were averaged for each sensor. The latter was then averaged on the thirteen sensors for each one of the ten RHs and reported in Figure 9 with the relative standard errors. The strongest response variation was in the low-humidity range (0 < RH < 20%), where it changed by ~80%, while at larger RHs (20% < RH < 40%), the response changed by just ~20%, as highlighted by the curve interpolating the data (Figure 9).

To assess the suitability of STN sensors to be employed in biological applications, a sample of 10 cc of Dulbecco Modified Eagle Medium, was measured three times in a row by the thirteen sensors. The average response (*n* = 39) was found to be 3.05±0.03, comparable with the sensor responses to 10 ppm of CO in dry conditions.

## 4. Discussion

STN sensors exhibited a very high statistical repeatability and reproducibility of their production process. These sensors also had an excellent sensitivity to a target gas, even in the presence of a high environmental RH percentage. To statistically evaluate their ability to discern low CO concentrations (from 4 to 1 ppm), the data confidence intervals for each RH (Figure 10) were calculated by Student’s t-test (confidence level = 0.95, α=0.05).

The resulting confidence intervals were well separated in presence of dry air only, and they were even more separated in the presence of humidity, making this sensor type particularly suitable for these applications where water vapor is unavoidable (although the responses were smaller with respect to dry air condition). In any case, STN sensor behavior, as all the MOX sensors, is significantly affected by humidity, that, if present in the environment before the gas target injection, leads to a non-negligible sensor response decrease to the latter (here CO). This mechanism is related to an unavoidable adsorption on the sensor film surface of $OH^{-}$ groups, originated by the water molecule dissociation (2) (at the WT of interest ranging from 300 to 600 °C), occupying most of the surface states [12,13].
(2)$$H_{2}O\to H^{+}+OH^{-}$$

To better understand the STN sensing behavior, it is crucial to consider the non-simple additivity of the responses of MOX sensors to two different gases (here CO and H_2_O), because of their power law behavior (see below). Therefore, the average responses of the thirteen sensors to different CO concentrations and to absolute humidities (AH, representing the mass of water vapor per mass of dry air, calculated as g/m^3^) were plotted as a 3D-surface (Figure 11) [25], which combines the outcomes reported in Figure 8 and Figure 9. CO concentration ranged between 10 and 1 ppm, while AH, calculated by using Clapeyron formula (Equation (3)), ranged between 0 and 2.5 g/m^3^:(3)AH(t)=RH(T(t))⋅A⋅e−BT(t), 
where $A=1.39\cdot 10^{8}kPa$ and B=5246K parameters are fitted with saturated water steam diagrams [31,32]. Humidity and CO, if separately injected, both affect the sensor conductance exponentially:(4)$$G_{H_{2}O}\;=\;a\;+\;bp_{H_{2}O}^{\alpha },\;$$
(5)$$G_{CO}\;=\;c\;+\;d{\left[CO\right]}^{\gamma },\;$$
where $p_{H_{2}O}$ is the water vapor pressure and the coefficients *a*, *b*, *c*, *d*, *α*, *γ* are constants, according with the sensor conductance models [33]. Considering that CO is injected at different $p_{H_{2}O}$, *c* and *d* are expected to depend by *b*. Instead, it was experimentally found that *α* and *γ* remain nearly constant (variations < 3%) at all the $p_{H_{2}O}$ tested. On this basis, combining (4) and (5), it is possible to describe the sensor conductance dependence by CO and water vapor concentrations as follows:(6)$$G_{p_{H_{2}O,\;\left[CO\right]}}=G_{0}+Ap_{H_{2}O}^{\alpha }+B{\left[CO\right]}^{\gamma }+Cp_{H_{2}O}^{\alpha }{\left[CO\right]}^{\gamma },\;$$
where $G_{0}\;$ is the sensor conductance [33] in the gas absence, and the constants *A*, *B*, *C*, *α*, *γ* can be inferred by fitting the experimental points according to (7). The average STN responses to the diverse CO concentrations at different RH percentages, transformed in absolute humidity degree (AH) with (3), were employed to calculate a 3D fitting surface (Figure 11).
(7)$$z=a+bx^{c}+dy^{e}+fx^{c}y^{e},\;$$

Since the responses started from zero, then a=0 in (7). Hence, the resulting 3D fitting surface is:(8)$$z=0.76x^{0.54}+0.33y^{0.60}-0.12x^{0.54}y^{0.60},$$

STN responses to CO is higher at small AH concentration (~0.5 g/m^3^) but become less evident at high AH values. In conclusion, STN sensors demonstrate, once again, suitability for application involving environments with a relatively high humidity degree (such as biological, medical, and environmental ones).

## 5. Conclusions

One of the main goals of the current research in the sensor field is to find out innovative devices to be employed in several diverse applications, such as environmental air monitoring, food safety assessment, pathologies detection, and so on. In particular, the use of sensors aimed to detect pathologies at their earliest stage allow to treat them as soon as possible, increasing the patient healing probability. To successfully employ these sensors in such devices, it is crucial to statistically test their production reproducibility, outcomes repeatability, and their behavior in the presence of gas and in unfavorable environmental conditions. The STN sensors employed in this study, heated at the WT of 500 °C, also showed a very high response repeatability in the presence of a high relative humidity degree (30–40%) and a very low CO concentration (up to 0.2 ppm) with almost negligible errors. The confidence intervals (Figure 10) confirmed the capability of the sensors to clearly and reliably discriminate between CO concentrations down to at least 1 ppm. Moreover, this discrimination was even higher in the presence of high relative humidity, although the response amplitude decreased as the latter increased, reaffirming the suitability of these sensors to be employed in relative wet conditions. The production process, chosen to synthesize the STN nanostructured film, showed a high reproducibility (with a maximum standard error of about 10%), allowing us to manufacture sensors with very similar behavior (as proved by the Arrhenius plots). These features, particularly the low interference of the water vapor in the STN sensors’ behavior, make them very promising for many diverse uses, e.g., in the medical, in-vitro biological applications, and in the environmental field, where humidity is always present.

Finally, these STN sensors can be completely characterized since the calibration surface constructed on the statistical outcomes computed on their responses can be represented by an exact mathematical equation.

Given the excellent performance exhibited by STN sensors in the CO detection (despite the presence of high humidity), the future aim is focused on the investigation of the STN behavior subjected to other gas types and interferers. Moreover, the STN nanostructures could be modified (changing the manufacture process parameters) to optimally detect any desired target gas.

## Figures and Tables

**Figure 1 sensors-23-01983-f001:**
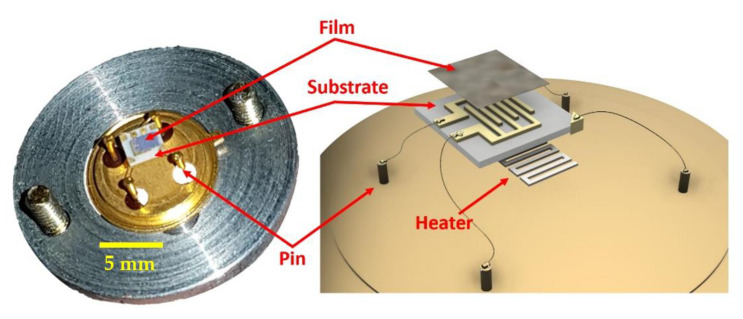
Sketch of a sensor. Left, photograph of a real sensor inserted in the fixing ring; right, three-dimensional drawing of the exploded view of a sensor.

**Figure 2 sensors-23-01983-f002:**
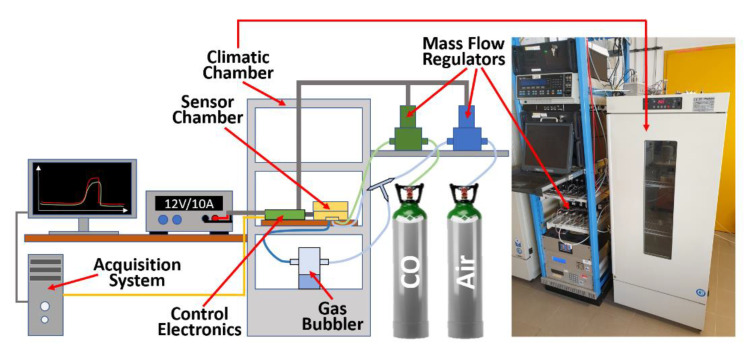
Sketch of the experimental set-up. Left, schematics of the sensor set-up: the air and CO (blue and green paths respectively) fluxes, precisely adjusted through two digital mass-flow regulators, are directed toward the sensor chamber; the air can also be directed inside a gas bubbler in order to obtain a desired humidity percentage. The control electronics, powered by a 12 V/10 A power supply, managed the digital mass-flow regulators, the sensor heating, and signal acquisition, allowing to monitor the experiments in real time. The acquired signals are sent to a computer (orange cable) through a serial port and plotted by custom software. Right, photograph of the actual set-up, sketched on the left.

**Figure 3 sensors-23-01983-f003:**
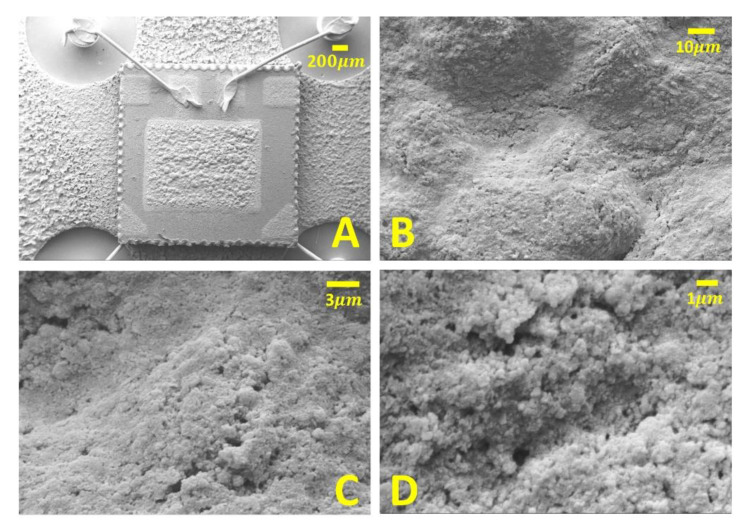
SEM micrographs of a gold-plated representative sensor. (**A**) Overview of the sensor connected to the TO-39 socket; (**B**–**D**) a selected region of the sensor viewed at progressively increasing magnitude. (**A**) Magnification = 70×, WD (working distance) = 11.0 mm, EHT (Electron High Tension) = 15.00 kV; (**B**) Magnification = 2.500×, WD = 11.0 mm, EHT = 15.00 kV; (**C**) Magnification = 10.000×, WD = 11.0 mm, EHT = 15.00 kV; (**D**) Magnification = 20.000×, WD = 9.5 mm, EHT = 15.00 kV.

**Figure 4 sensors-23-01983-f004:**
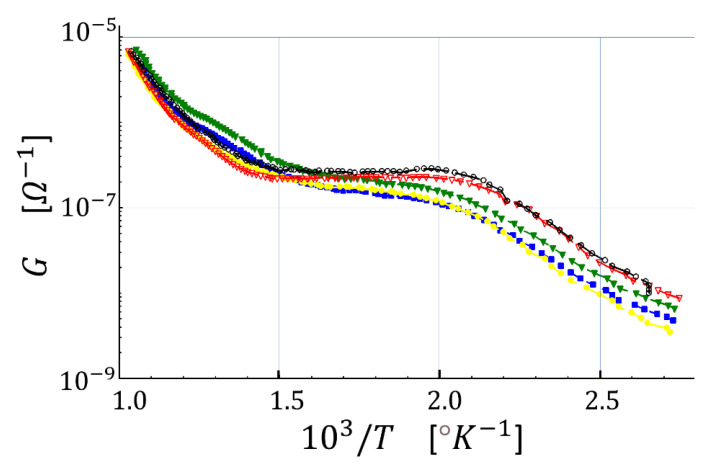
Arrhenius plot of five sensors in the presence of dry air. The sensor conductance (G) is plotted versus the reciprocal of the absolute temperature (T); the five colored lines represent the Arrhenius plots of the five representative sensors.

**Figure 5 sensors-23-01983-f005:**
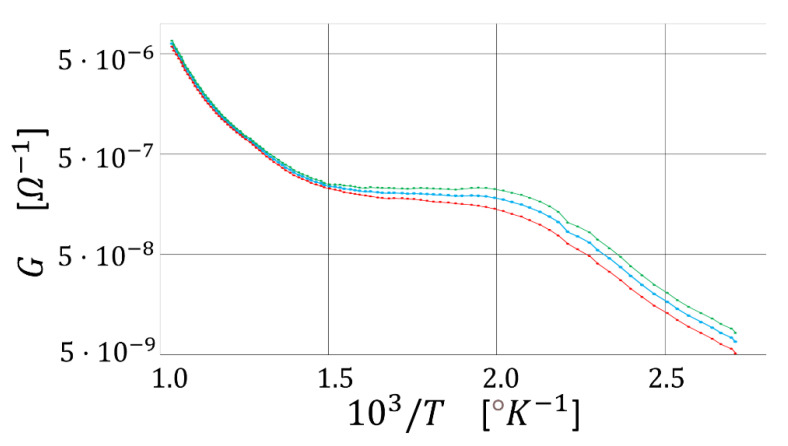
Average Arrhenius plot of the thirteen sensors. The average Arrhenius plot, the positive standard error, and the negative one are plotted as the blue, green, and red lines, respectively.

**Figure 6 sensors-23-01983-f006:**
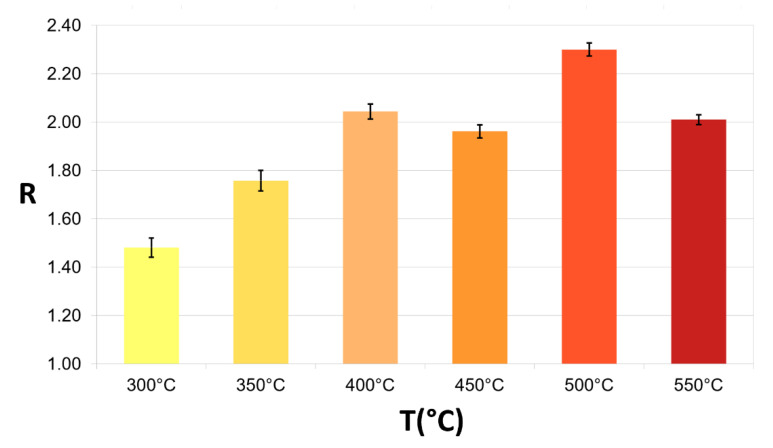
Bar graph of the average sensor responses with their standard error to CO at different WTs. The responses of the thirteen sensors, heated to different temperatures (from 300 °C to 550 °C, with discrete jumps of 50 °C), to 10 ppm CO were averaged (*n* = 39) and reported in the histogram with the relative standard errors.

**Figure 7 sensors-23-01983-f007:**
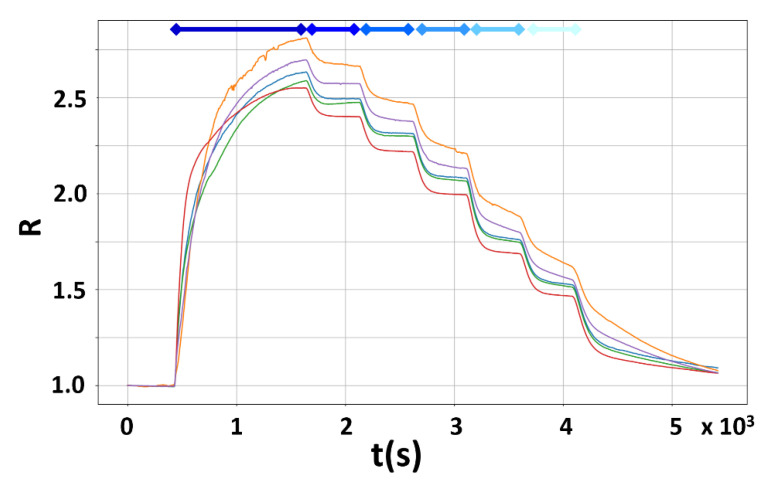
Sensor responses vs. time to decreasing CO concentrations in dry air. The superimposed responses of five representative sensors (red, green, blue, violet, and orange lines; the 1.0 response corresponds to zero CO concentration; Equation (1)) exposed to progressively decreasing CO concentrations (10, 8, 6, 4, 2, and 1 ppm, indicated by the progressively lightening blue bars) are plotted vs. time. Each CO exposition lasted 10 min.

**Figure 8 sensors-23-01983-f008:**
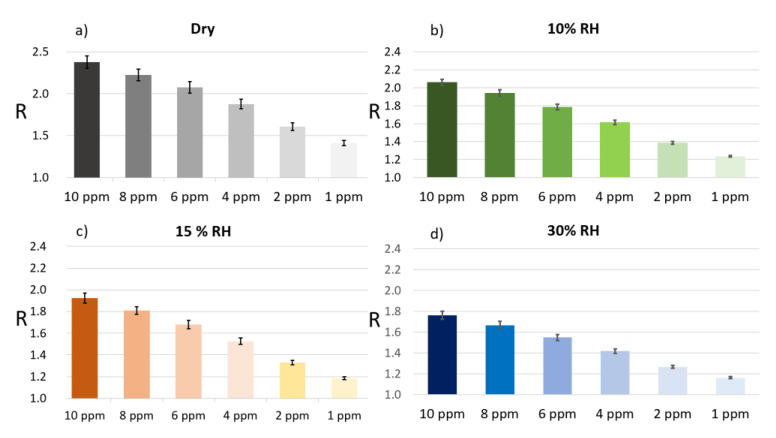
Average sensor responses and their standard errors to decreasing CO concentration and at different RH percentages. Each panel reports the average responses (*n* = 39) to different CO concentrations (10, 8, 6, 4, 2, and 1 ppm; progressively lightening candles) at a fixed humidity degree, that are: (**a**) dry conditions (grey), (**b**) RH = 10% (green), (**c**) RH = 15% (brown), and (**d**) RH = 30% (blue).

**Figure 9 sensors-23-01983-f009:**
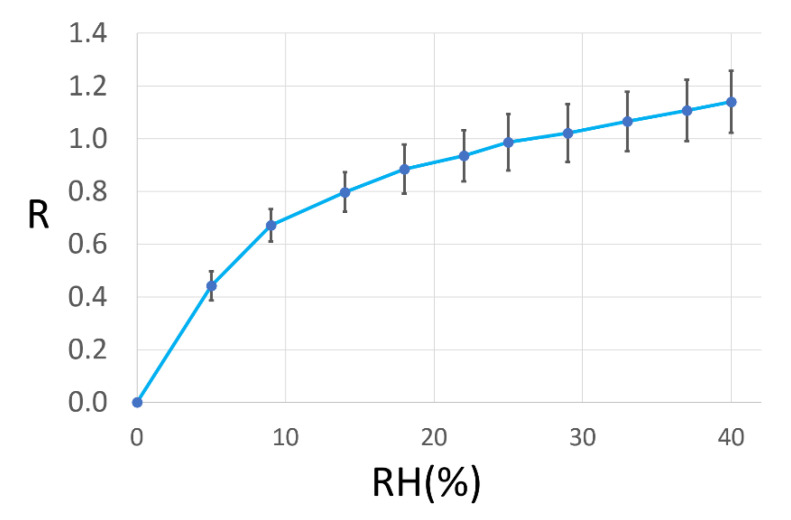
Average responses and standard errors of all the sensors to different RH percentages. Each data point represents the average response calculated on the thirteen sensors (*n* = 39) at a given RH (ranging from 0 to 40%) that are connected by the blue line for clarity; the zero-response corresponded to dry air (RH = 0%).

**Figure 10 sensors-23-01983-f010:**
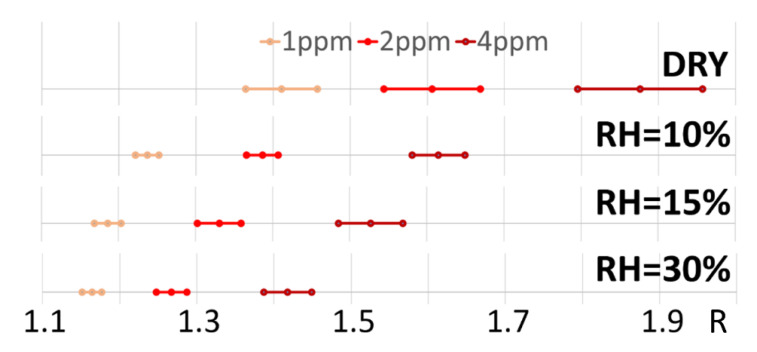
Confidence intervals for sensor responses at each RH percentage. The confidence intervals were calculated on the responses (*n* = 39; Figure 8) at 1, 2, and 4 ppm CO concentration (orange, light red, and dark red intervals, respectively), at 0, 10, 15, 30 % of RH.

**Figure 11 sensors-23-01983-f011:**
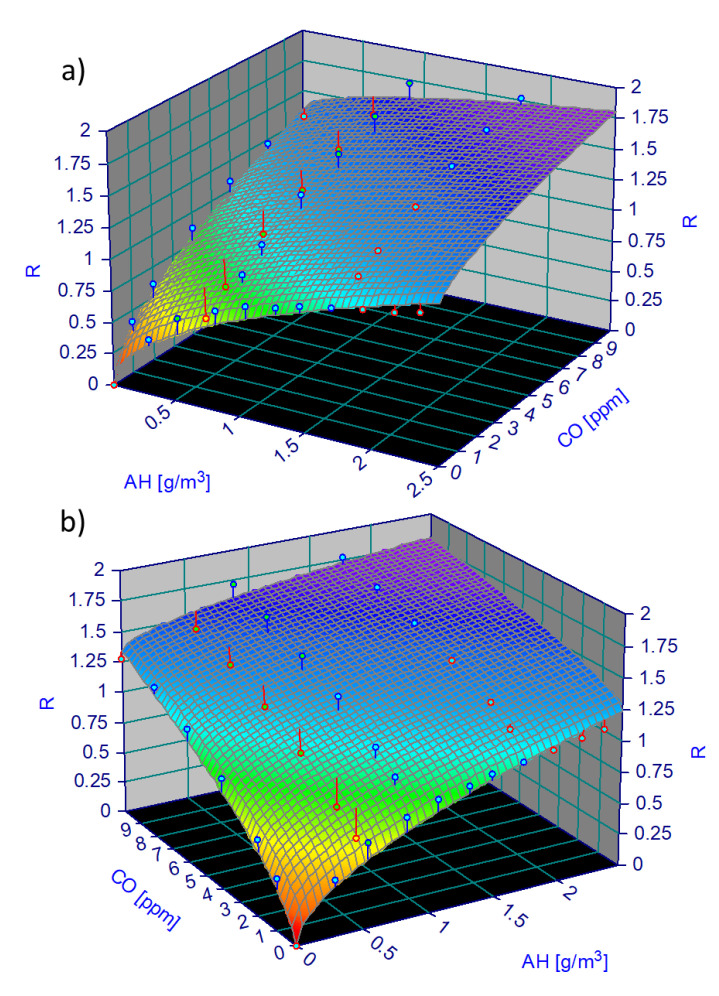
3-D calibration surface. Surface resulting by plotting the average responses (*n* = 13; z-axis) to CO (concentration range 0–10 ppm; x-axis) at different humidity conditions (0–2.5 g/m^3^; y-axis), shown with two different views (**a**,**b**)).

**Table 1 sensors-23-01983-t001:** Average sensor responses and related standard errors to CO at different WTs. The yellow, light orange, pink, orange, red, and dark red candles and table columns correspond to the 300, 350, 400, 450, 500, and 550 °C, respectively.

Data\T(°C)	300 °C	350 °C	400 °C	450 °C	500 °C	550 °C
Average	1.48	1.76	2.04	1.96	2.31	2.01
Std. Err.	0.04	0.04	0.03	0.03	0.03	0.02

## Data Availability

Not applicable.

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
