# Peer review of "Reproducibility and Repeatability Tests on (SnTiNb)O2 Sensors in Detecting ppm-Concentrations of CO and Up to 40% of Humidity: A Statistical Approach"

_sensors, 2023, doi:10.3390/s23041983_

Round 1

Reviewer 1 Report (Previous Reviewer 3)

Manuscript can be accepted

Author Response

We thank very much this Reviewer to have accepted our paper as it is.

Reviewer 2 Report (Previous Reviewer 4)

sensors-2184277

This manuscript investigated the reproducibility of the fabrication and the repeatability of the response of the (SnTiNb)O2 sensor. The results were analyzed with different methods in order to statistically evaluate their reproducibility and repeatability under different conditions. This version of manuscript is significantly improved compared to the previous version but there are several issues authors should take in account to improve the manuscript:

1-     It is necessary to clearly indicate the novelty of this paper.

2-     Please improve the English grammar and writing skills for this paper.

3-     Too many parentheses in an abstract are unusual. No need to define reproducibility and repeatability inside parentheses in the abstract.

4-     It is better to present different contents in the form of separate paragraphs in the introduction.

5-     Why did you choose only low humidity to investigate humidity?

6-     Morphology can change the optimum working temperature to MOX gas sensor. Discuss this fact and refer to the reference below: DOI: 10.1088/1361-6528/abfd54

7-     You mentioned the presence of VOCs in the introduction section, but there isn't any further explanation in the text body. Also, the placement of the word VOCs in the Keywords is strange to me.

8-     It is recommended to present a comparison table to confirm the novelty and superiority of the work.

9-     When people talk about the sensor, we care about 3Ss. Sensitivity, Stability and Selectivity.  Do you have any discussion about the stability and selectivity?

10-  It’s recommended to Reduce the number of author’s articles in ref [16-22] in the introduction.

Author Response

Please see the file in attachment.

Reviewer 3 Report (Previous Reviewer 2)

They answered my questions.

Author Response

We thank very much this Reviewer to have accepted our paper as it is.

Round 2

Reviewer 2 Report (Previous Reviewer 4)

N/A

This manuscript is a resubmission of an earlier submission. The following is a list of the peer review reports and author responses from that submission.

Round 1

Reviewer 1 Report

This is an interesting and good-quality original research paper (article) dealing with the study of reproducibility in the fabrication of selected thermo-activated thick-film (SnTiNb)O2 sensors obtained by the screen-printing deposition technology, together with the study of their repeatability in discrimination and detection of CO at the various working temperature and relative humidity. Moreover, on the base of deep analysis and discussion of obtained experimental results the precise specific gas sensors usable parameters have also been determined.

However, this manuscript contains some weak points, mainly concerning the structure of proposed information, that should be improved before the publication, as listed below:

-  A title of this paper does not precisely reflect its aim and scope, because it concerns the application of selected thermo-activated thick-film (SnTiNb)O2 sensors for discrimination and detection of CO at specific working conditions – it should be modified.

- Chapter2, subchapter 2.2 – Sensors structure - already contains the set of information on the procedure of determination of gas sensor response (R) curves together with the exemplary set of response curves (shown in Fig.2). It should be moved into the Chapter3, as the second new subchapter 3.2, where the most important gas sensor response experimental results are described.

- Chapter2, subchapter 2.3 – Scanning Electron Microscope analysis - already contains the set of SEM images (Fig.3) related to the selected gold-plated representative sensor. It should be moved into Chapter3, as the first subchapter, where the most important information on the morphology of selected gas sensors used in the performed studies. Instead of the above-mentioned information (and SEM images) in this subchapter 2.3 some obligatory basic information on SEM Instrument used in the performed experiments should be added.

- Chapter3 – in relation to the comments above, all the information currently proposed in subchapters 3.1-3.3 should be joined into one subchapter 3.2, where all the experimental results on CO gas sensors response at different experimental conditions (temperature, humidity) should be described being a base for their deep discussion, as proposed in the current Chapter 4.

-    Chapter 5 – Conclusions look slightly chaotic. It should be slightly modified, i.e, in the beginning, the aim and scope of the submitted paper should be mentioned. Then, the most important information obtained by the authors combined with their discussion for better recognition should be separately pointed out.

-   References - According to the Guide for Authors, the respective doi number should be optionally added at the end of every cited paper.

Reviewer 2 Report

Manuscript ID: sensors-2029628

Title: Reproducibility and Repeatability tests with tin-titanium-niobium oxide thick-film sensors

In this manuscript the sensor production process was done through the sol-gel technique and were tested with CO, for reproducibility and repeatability evaluation. Then the sensors heated at the optimal temperature and were subjected to different CO concentrations.

In my opinion this paper can be considered for publication after some revises.

1-    Equation 1 needs more explanation. Is this equation for only dry air or all conditions?

2-    In figure 2 the response time and recovery time were not calculated. From the figure these times are very large which are not suitable.

3-    The SEM images show homogeneity of the sensor film surface, but for sensor it is much better to have porous surface. How the authors explain about it?

4-    What is RH and AH explain about it.

5-    In figure 5, what is the label of vertical axis.

6-    In figure 5 the highest response is for dry conditions, but in caption of figure 6 is written “the zero-response corresponded to dry air”. Please clarify it.

7-    Does figure 8 has more information than the other figures? If yes explain about it.

Reviewer 3 Report

In this paper, thermo-activated thick-film (SnTiNb)O2 sensing material for gas sensing is discussed. The authors have not thoroughly elaborated on the topic and hence should be rejected .The comments are as follows

1. What do you mean by thick film sensors, Please read the thick film technology in detail. 

2. Sensor film fabrication needs revision, particularly, binders used during fabrication, annealing etc. No details are provided

3. Material characterisations such as XRD, TG/DTA and SEM etc of the synthesized material  are not detailed.

4. What about the selectivity of the sensor?

5. What is novelty of the work? 

6. Comparision with presently available gas sensors (operating temperature, sensitivity, response and recovery time, LOD etc is not discussed

7. References need tremendous revision.

Reviewer 4 Report

sensors-2029628

In this manuscript, authors investigated the reproducibility of the fabrication and the repeatability of the response of the (SnTiNb)O2 sensor. They tested the different operating temperatures for the sensors to find the optimum working temperature. Finally, the authors claim, sensors show good reproducibility. However, it's failed to find new contributions in the field. No new material/process/ properties/mechanism were found in this manuscript. The results are not outstanding. The discussion is quite simple. Therefore, I can't recommend for publication in a high-impact journal like Sensors.

Additional comments:

1-    The aim of the manuscript should be presented with more details at the end of introduction.

2-    The background literature review is very brief in the introduction. Show the importance of the work by adding appropriate and related references.

3-    Please mention the name and manufacturer of the SEM equipment.

4-    Add SEM photography conditions to the figure 3.

5-    SEM is not enough for characterization. More characterization techniques should be added. How are you sure of the correct growth? How are you sure of the absence of contamination? Various characterizations (XRD, UV visible, FTIR and etc.) can assure the reader that the correct deposition has occurred.

6-    The authors have described the gas sensing setup, but their description is not clear. I recommend providing a schematic of the setup and sensor. This article can be a good example: DOI: 10.3390/ma15041383.

7-    Please improve the English grammar and writing skills for this paper.

8-    Synthesis and film deposition: the name of materials and instruments should be presented with details.

9-    Abstract: The abstract should begin with a brief and precise statement of the problem, followed by a description of the study method and design, the major results, and the conclusions reached. But in this manuscript, the abstract is presented in a general mode.